# Talking about suicide: An uncontrolled trial of the effects of an Aboriginal and Torres Strait Islander mental health first aid program on knowledge, attitudes and intended and actual assisting actions

**Gregory Armstrong**[1]*, **Georgina Sutherland**[2], **Eliza Pross**[3], **Andrew Mackinnon**[4], **Nicola Reavley**[4], **Anthony F. Jorm**[4]

**1** Nossal Institute for Global Health, Melbourne School of Population and Global Health, The University of Melbourne, Melbourne, Victoria, Australia, **2** Centre for Health Equity, Melbourne School of Population and Global Health, The University of Melbourne, Melbourne, Victoria, Australia, **3** Mental Health First Aid Australia, Parkville, Victoria, Australia, **4** Centre for Mental Health, Melbourne School of Population and Global Health, The University of Melbourne, Melbourne, Victoria, Australia

* g.armstrong@unimelb.edu.au

**Data Availability Statement:** Data cannot be shared publicly. Participants were assured

## Abstract

### Objective

Suicide is a leading cause of death among Aboriginal and Torres Strait Islander people. Friends, family and frontline workers (for example, teachers, youth workers) are often best positioned to provide initial assistance if someone is at risk of suicide. We developed culturally appropriate expert consensus guidelines on how to provide mental health first aid to Australian Aboriginal and Torres Strait Islander people experiencing suicidal thoughts or behaviour and used this as the basis for a 5-hour suicide gatekeeper training course called Talking About Suicide. This paper describes the outcomes for participants in an uncontrolled trial of this training course.

### Methods

We undertook an uncontrolled trial of the Talking About Suicide course, delivered by Aboriginal and Torres Strait Islander Mental Health First Aid instructors to 192 adult (i.e. 18 years of age or older) Aboriginal and Torres Strait Islander (n = 110) and non-Indigenous (n = 82) participants. Questionnaires capturing self-report outcomes were self-administered immediately before (n = 192) and after attending the training course (n = 188), and at four-months follow-up (n = 98). Outcome measures were beliefs about suicide, stigmatising attitudes, confidence in ability to assist, and intended and actual actions to assist a suicidal person.

### Results

Despite a high level of suicide literacy among participants at pre-course measurement, improvements at post-course were observed in beliefs about suicide, stigmatising attitudes,

anonymity, which was also a feature of the consent form and plain language statement. This manuscript does not contain data relating to any particular individual participants. The plain language statement confirmed to participants that only aggregate data would be presented in publications. No consent was sought from participants to make the data publicly available, nor was it sought from the Aboriginal Community Controlled Health Organisations (ACCHOs) and other Aboriginal Corporations or community organisations, who gave permission for the training courses and associated data collection in their communities. Requests for restricted access to the underlying data can be sent to the Human Research Ethics Committee at The University of Melbourne (HREC 1646346.5) at the following email address: HumanEthics-Enquiries@unimelb. edu.au.

**Funding:** The study was funded by a National Health and Medical Research Council grant (#1076796) funded under the grant scheme "Mental Health Targeted Call for Research into Suicide Prevention in Aboriginal and Torres Strait Islander Youth". The lead author is funded by an Early Career Fellowship (GNT1138096) from the National Health and Medical Research Council in Australia. The funders had no role in study design, data collection and analysis, decision to publish, or preparation of the manuscript.

**Competing interests:** AFJ is unpaid Chair of the Board of Mental Health First Aid Australia, which is a not-for-profit organization.

confidence in ability to assist and intended assisting actions. While attrition at follow-up decreased statistical power, some improvements in beliefs about suicide, stigmatising attitudes and intended assisting actions remained statistically significant at follow-up. Importantly, actual assisting actions taken showed dramatic improvements between pre-course and follow-up. Participants reported feeling more confident to assist a suicidal person after the course and this was maintained at follow-up. The course was judged to be culturally appropriate by those participants who identified as Aboriginal and/or Torres Strait Islanders.

## Implications

The results of this uncontrolled trial were encouraging, suggesting that the Talking About Suicide course was able to improve participants' knowledge, attitudes, and intended assisting actions as well as actual actions taken.

## Background

Suicide is a leading cause of mortality for Aboriginal and Torres Strait Islander peoples in Australia, ranking in as the fifth leading cause of death in 2018 [1]. The suicide death rate for Aboriginal and Torres Strait Islander peoples is estimated to be 23.0 per 100,000, which is twice the rate for non-Indigenous Australians [1], and a large disparity also exists in the rates of suicidal ideation and attempts [2, 3]. Similarly high rates of Indigenous suicide are a distressing phenomenon in several other postcolonial countries, for example, Canada, the United States and New Zealand [4]. Suicide by Indigenous peoples worldwide is a complex socio-cultural, political, biological and psychological phenomenon that needs to be understood in the context of colonisation, loss of land and culture, trans-generational trauma, grief and loss, and racism and discrimination [5–7]. Additionally, the higher levels of social disadvantage experienced by many Indigenous peoples increases their risk of mental disorders, disability, substance abuse and a suite of stressful life events, for example, unemployment, homelessness, incarceration and family breakdown, all of which are well-documented suicide risk factors [8–14].

Despite the high level of need in many countries, there is a lack of documented and rigorously evaluated Indigenous suicide prevention programs [15, 16]. One recommended suicide prevention intervention is known as 'gatekeeper training' [17, 18], the underlying premise of which is that family, friends and frontline workers (for example, teachers, youth workers) are often best positioned to identify those at risk of suicidal behaviour and provide initial assistance. Gatekeeper training teaches groups of people in the community how to identify and support individuals who are at high risk of suicide and to refer them to appropriate community supports, including mental health services [19]. A systematic review of suicide prevention interventions targeting Indigenous populations in Australia, Canada, New Zealand and the United States found that gatekeeper training demonstrated encouraging results, with significant short-term increases in participant knowledge, skills, intentions to assist, and confidence in identifying and assisting individuals at risk of suicide [16].

In a similar vein to gatekeeper training, 'Mental Health First Aid' is defined as the assistance provided to a person developing a mental health problem, experiencing the worsening of an existing mental health problem or in a mental health crisis, until appropriate professional treatment is received or until the crisis resolves [20]. A meta-analysis of Mental Health First Aid interventions reported that they are associated with increases in mental health knowledge,

decreases in negative attitudes and increases in supportive behaviours towards people with mental health problems [21, 22]. In 2000, a Mental Health First Aid training program was established in Australia in response to the need for public education about mental illness and its treatment [23]. Later, an Aboriginal and Torres Strait Islander Mental Health First Aid (AMHFA) program was established [24]. Based on a series of expert-consensus guideline documents [25], the AMHFA program guides participants in how to provide initial assistance to an Aboriginal or Torres Strait Islander person with a mental health problem or in a mental health crisis, including depression, psychosis, substance use, or experiencing a traumatic event, a panic attack, suicidal thoughts or engaging in non-suicidal self-injury. AMHFA guidelines were also developed around '*Cultural Considerations and Communication Techniques*' and around '*Communicating with an Aboriginal or Torres Strait Islander Adolescent*' [25, 26], which represent over-arching guidance regardless of the mental health issue.

The AMHFA program is run through Mental Health First Aid Australia (MHFAA) who use a train-the-instructor model, whereby MHFAA train a pool of accredited AMHFA Instructors—who are all Aboriginal or Torres Strait Islander people—in how to deliver the course material to Aboriginal and Torres Strait Islander community members and non-Indigenous frontline workers in their respective communities, where they are already embedded and have local support. Over 500 people have trained as AMHFA Instructors across Australia, and they have delivered training courses to over 60,000 people in their communities. An initial 2009 evaluation of the AMHFA program based on data on the roll-out of the program and qualitative data obtained from focus group discussions found the program to be both culturally appropriate and acceptable to Aboriginal and Torres Strait Islander people [24]. A more recent uncontrolled trial of the AMHFA training course observed that training participants showed improved mental health literacy and confidence to assist [27].

As part of the process outlined above, AMHFA guidelines for assisting an Aboriginal or Torres Strait Islander person experiencing suicidal thoughts or suicidal behaviour were developed in 2009 [25]. These guidelines were recently updated [28, 29] and were subsequently used to develop the Talking About Suicide course (outlined below), an AMHFA brief suicide gatekeeper training course. This paper reports on an uncontrolled trial of the Talking About Suicide course, examining the effects of the training on beliefs about suicide, stigmatising attitudes, confidence in ability to assist, and intended and actual assisting actions.

## Methods

### Study design

The study was an uncontrolled trial of the effects of the Talking About Suicide course on both Aboriginal and Torres Strait Islander participants and non-Indigenous participants, given the training was designed to target both groups. Data were collected immediately before and after the completing the course and at four-months after the course (follow-up). The length of the follow-up period was a pragmatic decision based on funding. The research was approved by the Human Research Ethics Sub-Committee at the University of Melbourne (HREC No.1646346.5) and by the Aboriginal Health and Medical Research Council of New South Wales.

### Training course

The 5-hour Talking About Suicide course teaches people how to support an Aboriginal and/or Torres Strait Islander person who is experiencing suicidal thoughts. The course, delivered by accredited Aboriginal Mental Health First Aid Instructors upskilled in the new course material, was designed to be culturally appropriate for Aboriginal and/or Torres Strait Islander

communities while also being useful for non-Indigenous frontline workers in a position to assist an Aboriginal and/or Torres Strait Islander person who is experiencing suicidal thoughts. People attending the course learn about how to: 1) identify the risk factors and warning signs of suicide; 2) confidently support an Aboriginal and/or Torres Strait Islander person in crisis; 3) connect an Aboriginal and/or Torres Strait Islander person to appropriate professional assistance and to other cultural or community supports; and 4) manage their own self-care when assisting someone who is experiencing suicidal thoughts and behaviours.

The course acknowledges that the high rates of suicide in Aboriginal and Torres Strait Islander people today stem from the disrupting effects and systemic harms caused by colonisation and its aftermath, including loss of land and culture, trans-generational trauma, racism and discrimination. Strategies for providing support are underpinned by a holistic social and emotional wellbeing framework with an emphasis on the strengths of Aboriginal and Torres Strait Islander knowledge and support systems, diversity in cultural practices, languages, beliefs and ways of knowing and doing. As a nationally delivered course, the focus was on supporting first aiders and communities to identify local approaches and community strengths to support Aboriginal and Torres Strait Islander people with suicidal thoughts, recognising the diversity amongst people and communities. The guidance provided in the course was based on a recent Delphi expert consensus study in which a panel of 27 Aboriginal and Torres Strait Islander people with both professional and personal expertise in suicide prevention established best practice guidelines on how to provide mental health first aid to an Aboriginal or Torres Strait Islander person who is experiencing suicidal thoughts and behaviours [30].

The course includes face-to-face teaching including multimedia materials and interactive group activities (e.g. small and large group discussion, case study activities, planning for safety activities). An Aboriginal and Torres Strait Islander Project Reference Group played a central role in the development of the course, including reviewing cultural adaptation and acceptability of materials. These materials include scripted drama films demonstrating recommended assisting behaviours and videos of Aboriginal and Torres Strait Islander people with lived experience of suicidality talking about their experiences and what assisted them and Aboriginal and Torres Strait Islander artwork to illustrate helping actions. Each course participant is given a handbook [31] that presents facts about suicide in Australia, details how to implement the recommended actions when assisting a suicidal person, and provides additional information on useful resources. Additional information about the course can be found at: https://mhfa.com.au/courses/public/types/amhfa-talking-about-suicide

## Participants and recruitment

Participants were adults (i.e. 18 years of age or older) who were either Aboriginal and/or Torres Strait Islander or non-Indigenous people who were frontline health and community services workers who worked with Aboriginal and Torres Strait Islander people. Some additional participants were Aboriginal and Torres Strait Islander community members wanting to learn how to better support their peers.

Participants attended one of 15 courses run in 13 metropolitan, rural and remote communities along the east coast of Queensland, Victoria and New South Wales. Communities were selected on the basis of pre-existing relationships between researchers, AMHFA Instructors and communities, and where Aboriginal and Torres Strait Islander suicide prevention was considered to be a high priority by the community. Courses were run in close partnership with Aboriginal Community Controlled Health Organisations (ACCHOs) and other Aboriginal Corporations or community organisations, who gave permission for the training courses and associated data collection in their communities. In partnership with Aboriginal and Torres

Islander researchers, Instructors and community-controlled organisations, the recruitment process harnessed the relational nature of Aboriginal and Torres Strait Islander communities. Advertisements for participation in the courses were distributed through the networks of the local ACCHO's and other community organisations, and those of the AMHFA Instructor. As a result, attendees self-selected to attend the training and participate in the study and were a diverse group of people connected to the local ACCHO and others from the broader community. Everyone who attended was invited to participate in the research study, although participation was not mandatory and anyone who did not wish to participate was still included in the course. Typically, courses were run at the office of the local ACCHO or at a hired venue that was known, culturally safe and accessible to participants, with the local ACCHO also acting as an additional source of support after the course.

## Outcome measures

Data were collected using a self-administered questionnaire comprising questions regarding participant sociodemographic characteristics and a range of outcome measures: beliefs about suicide, stigmatising attitudes, confidence in ability to assist, and intended and actual assisting actions. *Beliefs about suicide* were assessed by asking participants to agree or disagree on a 5-point Likert scale (strongly agree to strongly disagree) with each of five common suicide myths that are directly addressed in the training course [32–34].

Responses to a vignette were used to measure stigmatising attitudes, confidence in ability to assist and intended assisting actions. The vignette (see Fig 1) describes a 20-year-old male called Bala who, in the context of a potential relationship breakdown, is withdrawing from his friends, feeling worthless, increasing his alcohol use, and making statements about how he doesn't see the point anymore. *Stigmatising attitudes* were assessed using the Personal Stigma Scale [35] which asks participants whether they agree or disagree on a 5-point Likert scale (strongly agree to strongly disagree, with lower scores indicating lower stigma) with nine statements related to someone like Bala that sit on one of two sub-scales; "weak-not-sick" comprised of three items (e.g. 'a problem like Bala's is a sign of personal weakness') and "dangerous-unpredictable" comprised of six items (e.g. 'people with a problem like Bala's are unpredictable'). On recommendation from our Aboriginal and Torres Strait Islander Project Reference Group, a tenth statement was added to the inventory, 'people with a problem like Bala's just need a bit of tough love'. A factor analysis of pre-course responses found that the 'tough love' item loaded substantially (loading = 0.67) with the original items on the 'Weak not sick' scale on a single factor model (range 0.59–0.78). Adding the 'tough love' item to the 'Weak not sick' items increased this scale's Cronbach's alpha from 0.76 to 0.80. Items

Bala is 20 years old. He has been feeling unusually irritable for the past few weeks and has become distant with his family. Bala used to be a social drinker, but he is drinking more than usual and often on his own. He has lost interest in hanging out with his friends and has said that he feels like no one understands him. He has been fighting with his girlfriend a lot more over the past few months and their relationship has been 'on again, off again'. She is saying that she wants to leave him for good this time. Bala is feeling worthless and is making statements about how he doesn't see the point anymore. He keeps saying that he'll have nothing left if she leaves him. Bala has been thinking about how his cousin hung himself almost a year ago and has been talking a lot about death and suicide.

**Fig 1. Vignette in questionnaire.**

comprising the 'Dangerous/' scale also had acceptable loadings on a single factor model (range 0.30–0.75) and had a Cronbach's alpha of 0.71.

*Confidence in ability to assist* was assessed by asking participants how confident they were in assisting Bala, with response options on a five-point Likert scale from not at all confident to extremely confident [36, 37]. *Intended assisting actions* were assessed by asking participants how likely they were to do one of 15 actions on a five-point Likert scale from very unlikely to very likely. The actions were a mix of 11 actions that were explicitly recommended and four actions that were explicitly recommended to be avoided, based on the above-mentioned expert consensus guidelines we developed on how to provide mental health first aid to an Aboriginal or Torres Strait Islander person who is experiencing suicidal thoughts [30]. Changes in each action from before the course to immediately afterwards and four months later were examined for each action individually and scales were constructed. Separate scales of 'recommended' and 'not recommended/contrary' actions were created due to the potential for different patterns of change. Cronbach's alpha for the recommended actions scale was 0.86 while alpha for the 'not recommended/contrary' actions scale was only 0.44. The low value for the latter scale may partially reflect its comprising only four items.

*Actual assisting actions* were assessed at pre-course and follow-up by asking participants whether they had spoken with one or more Aboriginal or Torres Strait Islander people who they were concerned may be having suicidal thoughts. The timeframes were the 12 months prior to the course and the 4 months prior to completing the follow-up assessment. Recalling the person they had supported the most (if there was more than one), participants were asked which of a list of 17 actions they had taken to support that person. The list comprised of 13 actions that are explicitly recommended and four actions that are explicitly recommended to be avoided in the expert consensus guidelines [30]. For both the intended and actual assisting actions, the sequence of recommended and not recommended actions was deliberately mixed in the inventory. Responses were binary–yes/no–rather than the Likert scale used in response to the vignette.

Participants were also asked purpose-designed questions using four-point Likert scales to capture perceptions of the relevance of the course (not at all relevant to very relevant), how new the information was (not at all new to very new), how well it was presented (not at all well to very well) and how culturally appropriate the advice provided during the course was (not at all appropriate to very appropriate). Finally, we had anticipated that the course content may be a source of distress for some people, so we asked two purpose designed questions to assess whether people felt distressed (e.g. sad, stressed, overwhelmed, etc) about any aspects of the course and whether participants were glad to have attended the training course despite the fact that talking about suicide can cause distress for some people.

## Sample size

The target sample size was 170 participants. This allowed for the detection of small to medium effects (d = 0.25) in the total sample assuming a correlation of 0.50 between pre-course and subsequent occasions and allowing for approximately 25% attrition (StataCorp, 2015).

## Statistical analyses

All analyses were undertaken using Stata 14.2 [38]. Mean changes between time points were assessed using linear mixed model repeated measures (MMRMs) ANOVA with an unstructured variance-covariance matrix. Degrees of freedom were estimated using the Kenward-Rogers method [39]. Where scales were formed from multiple items, missing responses were imputed as mean values when a respondent had answered at least 75% of the items on the

scale. Many of the variables evaluated had skewed distributions that were likely to yield skewed residuals, formally violating the model assumptions. Transforming scores was judged unlikely to be successful in dealing with these problems. Accordingly, this problem was addressed when necessary using bootstrapping and calculation of bias-corrected parameter confidence intervals to assess the robustness of conclusions reached using conventional methods. This approach was used in preference to generalized (non-linear) modelling as it yields parameters that can be easily interpreted in terms of mean change rather than likelihood of responding in higher categories, as would be case for ordinal or count data models. Effect sizes were calculated using Glass's delta [40]. This variation of Cohen d statistic used the pre-course standard deviation as the basis of standardisation. Values of delta reported can be interpreted as the extent of mean change induced by the course within the context of the original distribution of the variable concerned, with the size of the effect indicated by the following conventions; 0.3 for small, 0.5 for medium and 0.7 for large.

For the analysis of repeated binary responses in relation to actual assisting actions, a mixed effect Poisson model was used, including an exposure term to accommodate the different periods of reporting. This allows inclusion of participants who reported assisting others either before the course, at follow-up or at both time points. Unlike a logistic model, the parameters yielded reflect the relative likelihood of taking an action (rather than the odds ratio). This is appropriate when the events of interest are prevalent. Robust standard errors were used as is appropriate when the 'count' data are binary [41].

## Results

### Participant characteristics

We recruited and obtained pre-course data from 192 participants. Most participants were retained at post-course measurement (n = 188, 98%), and 98 (51%) were retained at four-month follow-up. Table 1 shows the participant characteristics. Their mean age was 38.6 years and over three quarters were female. A little over half (57.3%) the participants identified as Aboriginal and/or Torres Strait Islander. The majority of participants (70.6%) lived outside capital cities. Most had at least completed secondary school and 40% had an undergraduate degree or higher, while a minority (13%) had completed year 10 or less. The majority of participants (81.1%) worked in 'frontline' roles delivering a variety of health and/or community services directly to the public. For 84.3% of participants, they had frontline work where some or all of their clients were of an Aboriginal or Torres Strait Islander background. Almost half (47.4%) had previously undertaken some form of MHFA training and 53.7% had undertaken other forms mental health training. A slightly lower proportion (42.8%) reported prior training in supporting a person having suicidal thoughts. All but one participant reported personal (i.e. self, family, friends or community) or workplace experiences of suicidality or death from suicide. Suicidality was particularly notable in family, clients and friends, while experience of suicide deaths was notable in friends, family and the broader community.

### Analysis of dropout

As only a little over half of the participants responded at the four-month follow-up, an analysis was undertaken exploring factors differentiating those who participated at follow-up from those who did not. Few of the predictors of missingness were strong in isolation and basic demographic factors including age, gender and educational achievement were not associated with missingness at follow-up. However, it is likely that individuals who identified as Aboriginal or Torres Strait Islander or who lived in remote areas, and participants who had less suicide prevention knowledge and expertise at baseline may be under-represented in follow-up data.

**Table 1. Participant characteristics (n = 192).**

|  | N(%) |
|---|---|
| Age–Mean years (SD) | 38.6 (12.4) |
| Gender |  |
| Male | 29 (15.1%) |
| Female | 163 (84.9%) |
| Aboriginal and/or Torres Strait Islander Identification |  |
| Aboriginal | 91 (47.4%) |
| Torres Strait Islander | 9 (4.7%) |
| Aboriginal and Torres Strait Islander | 10 (5.2%) |
| Neither | 82 (42.7%) |
| Place of Residence |  |
| Capital city | 53 (29.4%) |
| Regional city | 65 (36.1%) |
| Rural area | 50 (27.8%) |
| Remote area | 12 (6.7%) |
| Education |  |
| Year 9 or lower | 6 (3.2%) |
| Completed Year 10 | 18 (9.5%) |
| Completed Year 12 | 18 (9.5%) |
| Trade certificate/apprenticeship | 5 (2.6%) |
| Other certificate | 49 (25.5%) |
| Associate or undergrad diploma | 20 (10.5%) |
| Bachelor's degree or higher | 76 (40.0%) |
| Employment[†] |  |
| Law enforcement | 4 (2.3%) |
| Centrelink | 7 (4%) |
| Legal Services | 7 (4%) |
| Aged care | 11 (6.3%) |
| Financial counselling | 12 (6.9%) |
| Disability | 13 (7.4%) |
| Employment services | 13 (7.4%) |
| Community housing and homelessness | 15 (8.6%) |
| Drug and alcohol | 16 (9.1%) |
| Mental health | 29 (16.6%) |
| Other health services | 33 (18.9%) |
| Education | 50 (28.6%) |
| Other employment | 52 (29.7%) |
| Not a 'frontline worker' | 33 (18.9%) |
| Clients ATSI people |  |
| Some of clients ATSI | 79 (45.9%) |
| All clients ATSI | 66 (38.4%) |
| No | 27 (15.7%) |
| Previous MHFA training |  |
| Yes, less than 5 years | 77 (40.1%) |
| Yes, 5 or more years ago | 14 (7.3%) |
| No or 'not sure' | 101 (52.6%) |
| Other mental health training |  |
| Short course(s) | 54 (28.4%) |

*(Continued)*

**Table 1.** (Continued)

|  | N(%) |
|---|---|
| A mental health subject in a course | 27 (14.2%) |
| Whole certificate/diploma/degree | 15 (7.9%) |
| Other | 6 (3.2%) |
| No or 'not sure' | 88 (46.3%) |
| Training in support for suicidal persons |  |
| Yes | 96 (42.2%) |
| No or 'not sure' | 111 (57.8%) |
| Personal experience of suicidality[†] |  |
| Self | 24 (12.6%) |
| Family | 75 (39.3%) |
| Friends | 57 (29.8%) |
| Colleagues | 14 (7.3%) |
| Clients/patients | 78 (40.8%) |
| Broader community network | 38 (19.9%) |
| None of the above | 24 (12.6%) |
| I'd rather not say | 17 (8.9%) |
| Personal experience of suicide death[†] |  |
| Family | 78 (41.5%) |
| Friends | 104 (55.3%) |
| Colleagues | 19 (10.1%) |
| Clients/patients | 31 (16.5%) |
| Broader community network | 80 (42.6%) |
| None of the above | 18 (9.6%) |
| Do not wish to say | 6 (3.2%) |

† Multiple responses permitted.

## Beliefs about suicide

Most participants held evidence-consistent beliefs before the training. Before taking the course, between 71.2% and 91.6% of participants responded either 'Disagree' or 'Strongly Disagree'–evidence consistent responses–to four of the five myth statements. Although these specific belief statements weren't discussed directly in the course, participants still appear to have gained enough knowledge to influence their attitudes towards suicide myths. With little room to improve, the greatest changes seen after the course were from 'Disagree' or 'Strongly Disagree'–reflecting greater certainty about these positions. Responses to the item 'You can tell how serious someone is about suicide by the method they are thinking of using' deviated from this pattern, with just 29.8% of participants strongly disagreeing with this belief and 14.9% disagreeing (a total of 44.8%); after the course, evidence-based responses were reported by over half the participants (56.1%) but one third (33.1%) continued to believe that the method of suicide contemplated was indicative of seriousness.

Mean item responses on each occasion of measurement are shown in Table 2. All changes from pre-course values to post-course values were statistically significant, with the exception of the item concerning the seriousness of threats made under the influence of alcohol and other drugs. Effect sizes ranged from values that would be considered small to large size effects, except for the alcohol/drugs item. At follow-up, means generally fell back toward pre-course values becoming non-significant compared to this time. This trend was also observed for the

**Table 2. Level of disagreement with myths about suicide.**

| Belief Item | Pre-course (n = 190)‡ | | Post-course (n = 189) ‡ | | | Follow-up (n = 97) ‡ | | |
|---|---|---|---|---|---|---|---|---|
| | Mean | SD | Mean | SD | Δ† | Mean | SD | Δ† |
| Asking about suicide can put the idea into someone's head | 3.88 | 0.88 | 4.62*** | 0.73 | 0.84 | 4.22* | 0.94 | 0.38 |
| If a person is talking about killing themselves then there is nothing you can do to stop them. | 4.38 | 0.79 | 4.60*** | 0.75 | 0.28 | 4.42 | 0.88 | 0.06 |
| Suicidal threats made under the influence of alcohol and other drugs do not need to be taken seriously. | 4.32 | 1.12 | 4.35 | 1.27 | 0.02 | 4.51 | 1.03 | 0.16 |
| You can tell how serious someone is about suicide by the method they are thinking of using. | 3.18 | 1.21 | 3.50** | 1.44 | 0.27 | 3.43 | 1.38 | 0.21 |
| All people who are suicidal want to die | 4.09 | 0.87 | 4.25* | 0.98 | 0.18 | 4.27 | 0.91 | 0.21 |

NB: Higher scores are more consistent with evidence-based responses.

† Glass's Delta compared to pre-course mean using pre-course standard deviation.

‡ Number of observations varies slightly due to missing data.

* = p<0.05

** = p<0.01

*** = p<0.001.

belief that asking about suicide can put the idea into a person's head, but the follow-up mean remained significantly above its pre-course value.

## Stigmatising attitudes

Mean changes in stigmatising attitudes on the Personal Stigma Scale are reported in Table 3. The distribution of scores for the sub-scale 'Weak-not-sick' were highly skewed, with 32.3% of participants having the minimum possible score (5) before the course. Nevertheless, there were reductions in scores after the course, with nearly half of participants (49.7%) recording the lowest possible score. This change was maintained at follow-up when 48.3% of participants recorded the lowest possible score. Analyses of means found that changes in the 'Weak-not-sick' scale from pre-course means were significant after the course but not at follow-up ($t(132.2) = 4.27$, $p<0.0001$; and $t(185.9) = 1.23$, $p = 0.2214$, respectively). Probably reflecting a floor effect, the changes from pre-course were small to below medium size.

In contrast to the Weak-not-sick score distributions, those of the Dangerous/Unpredictable sub-scale were much less skewed, with a modal response of 12 (corresponding to an average 'disagree' response). After the course the modal response was 6 –the lowest possible score. Changes in means (see Table 5) from pre-course means were significant both post-course and at follow-up ($t(185.2) = 7.51$, $p<0.0001$; and $t(126.2) = 3.07$, $p = .0026$, respectively). The changes from pre-course were medium in size.

**Table 3. Stigmatising attitudes.**

| Item | Pre course (n = 192) | | Post course (n = 185) | | | Follow-up (n = 89) | | |
|---|---|---|---|---|---|---|---|---|
| | Mean | SD | Mean | SD | Δ† | Mean | SD | Δ† |
| 'Weak not sick' | 7.77 | 2.81 | 7.04*** | 2.88 | 0.25 | 6.91 | 2.48 | 0.30 |
| 'Dangerous/' | 12.20 | 3.35 | 10.70*** | 3.60 | 0.45 | 10.76** | 3.24 | 0.43 |

† Glass's Delta compared to pre-course mean using pre-course standard deviation.

* = p<0.05

** = p<0.01

*** = p<0.001.

## Confidence in ability to assist

The modal response to the question asking participants about their confidence to assist Bala moved from a modal response of 'Moderately confident' to 'Quite a bit confident' after the course and at follow-up. The proportion of participants who felt 'not at all confident' or 'only a little bit confident' fell from a quarter (29.3%) to just 3.4% after the course. Consistent with the pattern of endorsement, mean responses increased from 3.12 (SD 1.00) before the course to 4.11 (SD 0.78) afterward and 3.92 (SD 0.82) at follow-up. The changes correspond to large effects sizes ($\Delta = 0.98$ and $\Delta = 0.80$, respectively). Change from pre-course means at both times was statistically significant (t(187.0) = 14.74, p<0.0001; and t(124.9) = 9.56, p<0.0001, respectively).

## Intended assisting actions

**Intentions related to recommended assisting actions.** Before the course, most participants responded that they would be likely or very likely to implement most of the recommended actions. Endorsement rates for six actions were above 90%–some nearly universal–with four others being in the range of 70% to 80%. Asking about a plan was an outlier, with only 58.2% of likely or very likely responses. Given the high rates of endorsement, large changes could not be observed for many recommended actions. However even under these circumstances, increases in the proportion of participants responding 'very likely' rather than 'likely' after the course led to endorsement rates uniformly above 95% for all actions.

Formal analyses of change in mean responses using MMRMs were statistically significant for all actions after the course (see Table 4). Mean changes at follow-up were significant for only five actions. These included listening, asking about suicide, asking about a plan, asking about previous attempts, and calling for support if the participant felt they were out of their

**Table 4. Intentions to implement recommended helping actions.**

| Action | Pre course (n = 192)[‡] | | Post course (n = 188)[‡] | | | Follow-up (n = 91)[‡] | | |
|---|---|---|---|---|---|---|---|---|
| | Mean | SD | Mean | SD | $\Delta$[†] | Mean | SD | $\Delta$[†] |
| Spend time listening to Bala discuss his feelings | 4.57 | 0.79 | 4.81*** | 0.60 | 0.30 | 4.79* | 0.55 | 0.28 |
| Ask Bala directly about suicidal thoughts | 3.86 | 0.94 | 4.73*** | 0.53 | 0.92 | 4.58*** | 0.72 | 0.76 |
| Ask Bala directly if he has a plan for how he would kill himself | 3.57 | 1.26 | 4.66*** | 0.62 | 0.87 | 4.39*** | 0.87 | 0.65 |
| Ask Bala directly if he has attempted suicide in the past | 4.06 | 0.89 | 4.70*** | 0.47 | 0.72 | 4.47*** | 0.74 | 0.46 |
| Allow Bala to discuss his reasons for wanting to die | 4.30 | 0.74 | 4.73*** | 0.46 | 0.59 | 4.48 | 0.72 | 0.25 |
| Ask Bala how he would like to be supported | 4.61 | 0.56 | 4.80*** | 0.44 | 0.34 | 4.65 | 0.66 | 0.08 |
| Ask Bala whether there is someone he would trust to help support him | 4.64 | 0.54 | 4.84*** | 0.37 | 0.36 | 4.69 | 0.64 | 0.09 |
| Encourage Bala to get professional help as soon as possible | 4.60 | 0.66 | 4.82*** | 0.44 | 0.34 | 4.71 | 0.75 | 0.18 |
| Provide Bala with information about where he can seek help | 4.66 | 0.61 | 4.85*** | 0.36 | 0.30 | 4.79 | 0.61 | 0.22 |
| Seek Bala's permission to contact his regular doctor or mental health professional about his thoughts of suicide | 4.21 | 0.97 | 4.65*** | 0.69 | 0.45 | 4.33 | 0.94 | 0.12 |
| Call emergency or mental health services if you are concerned and you feel out of your depth | 4.43 | 0.78 | 4.83*** | 0.40 | 0.51 | 4.74*** | 0.63 | 0.39 |
| Scale (average) | 4.32 | 0.53 | 4.76*** | 0.33 | 0.83 | 4.60** | 0.57 | 0.53 |

‡ Numbers vary slightly between actions due to missing responses.

†: Glass's Delta compared to pre-course mean using pre-course standard deviation.

* = p<0.05

** = p<0.01

*** = p<0.001.

**Table 5. Intentions to implement non-recommended helping actions.**

| Action | Pre course (n = 192)[‡] | | Post course (n = 188) [‡] | | | Follow-up (n = 91) [‡] | | |
|---|---|---|---|---|---|---|---|---|
| | Mean | SD | Mean | SD | Δ[†] | Mean | SD | Δ[†] |
| Wait and see if things get worse before doing anything. | 1.71 | 0.88 | 1.40*** | 0.85 | 0.34 | 1.61 | 0.97 | 0.11 |
| Promise Bala that you will not tell anyone about his suicidal thoughts. | 2.40 | 1.17 | 1.61*** | 0.95 | 0.68 | 1.75*** | 1.01 | 0.56 |
| Try to convince Bala that suicide is the wrong thing to do. | 4.22 | 0.89 | 4.30 | 1.03 | -0.09 | 4.22 | 0.98 | 0.00 |
| Take charge of the situation for Bala. | 2.98 | 1.06 | 3.42*** | 1.38 | -0.41 | 3.16** | 1.19 | -0.17 |

‡ Numbers vary slightly between actions due to missing responses.

†: Glass's Delta compared to pre-course mean using pre-course standard deviation.

* = $p < 0.05$

** = $p < 0.01$

*** = $p < 0.001$.

depth. All means at follow-up remained higher than before the course, representing improvement, albeit not statistically significant. A scale of recommended actions comprising all nine items had a Cronbach's alpha of 0.86 assessed pre-course. Reflecting the pattern of individual action items, the increase in likelihood of taking recommended actions was significant after the course and also at follow-up ($t(190.5) = 11.87$, $p < .0001$ and $t(116.9) = 3.47$, $p = 0.0007$, respectively). The changes from before the course to after the course were of large size (i.e. 0.83) and remained medium sized at follow-up (i.e. 0.53), although they ranged widely across individual intended actions at both post-course (range:0.30–0.92) and follow-up (range:0.08–0.76).

**Intentions related to non-recommended actions.** Analysis of change in mean responses to each item using MMRM showed statistically significant reductions (the desired direction for actions not recommended) after the course for two of the four actions (see Table 5). Conversely, there was no change in the likelihood of trying to convince Bala not to act and a significant increase for taking charge of the situation. The reduction in promising not to tell anyone remained relatively stable and significant as did the increase for taking charge of the situation. The changes from pre-course were medium to small in size.

## Actual assisting actions

Well over half of participants (58.6%; 112/191) reported talking with a suicidal person in the 12 months prior to the course. Despite the shorter time frame of the question posed at follow-up, a substantial percentage (44.4%, 40/91) reported assisting someone with suicidal thoughts in the four months between the course and follow-up data collection.

After inclusion of an exposure term to accommodate the different length of reporting periods, we estimated the rate of assisting at follow-up to be over twice that prior to the course (incident rate ratio = 2.58, 95%CI: 2.14–3.11). As exposure was completely confounded with occasion of reporting in these data it was not possible to account for the different reporting periods and within-person variation in the same model. This could inflate significance tests and reduce confidence intervals. The analysis also relies on there being comparable accuracy of reporting events over the 12-month pre-course and 4-month post-course periods. Thus, the higher rate estimate should be treated with caution.

Fig 2A and 2B show the percentage of participants taking each of the actual recommended and non-recommended assisting actions in each time period. It shows dramatic increases in rates of taking many recommended actions in the follow-up period compared to before the

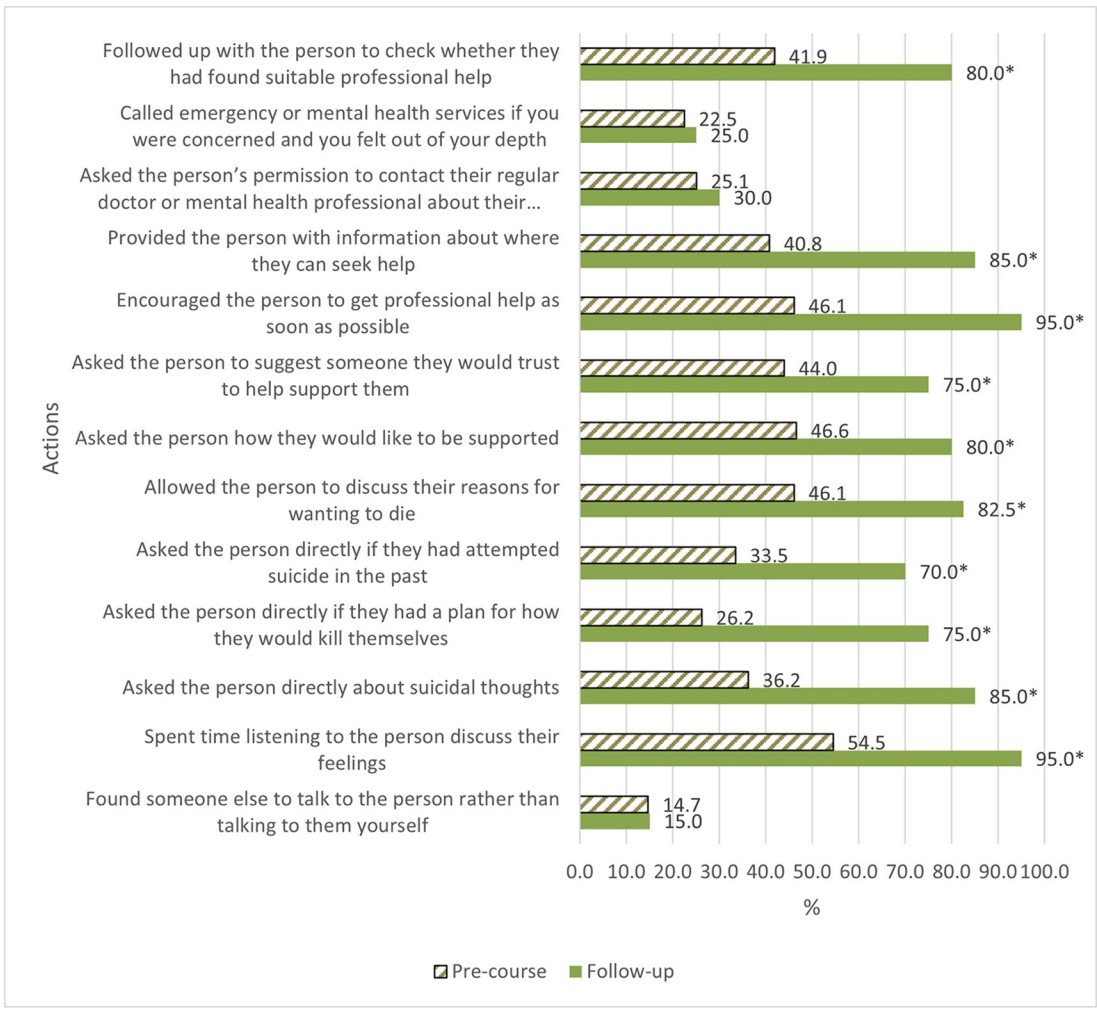

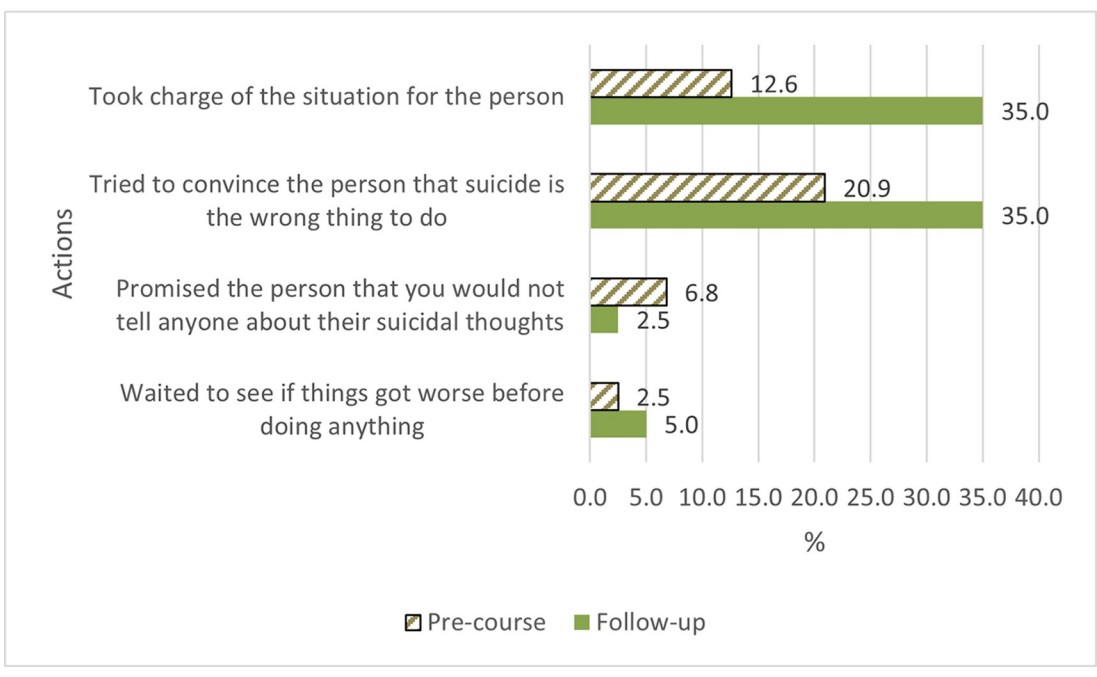

**Fig 2.** A. Percentage of participants taking particular recommended actions talking to a suicidal Aboriginal or Torres Strait Islander person before the course (n = 191) and at follow-up (n = 40). B. Percentage of participants taking particular non-recommended actions talking to a suicidal Aboriginal or Torres Strait Islander person before the course (n = 191) and at follow-up (n = 40). * p<0.05.

course. Recommended actions that did not increase substantially or statistically significantly were those related to referring the person directly to another source of assistance. Among actions that were recommended to be avoided, 'waiting and seeing' and 'promising not to tell' were reported as actions that were taken infrequently prior to the course and no statistically change in their use were observed. Taking charge of the situation and trying to convince the person that suicide is the wrong thing to do—both non-recommended actions—showed increases in use after the course although these changes did not reach statistical significance (OR = 5.67; 95% CI: 0.93–34.47; p = 0.059, and OR = 2.29; 95% CI: 0.89–5.89; p = 0.086).

## Participant satisfaction of the course

The overwhelming majority of participants were happy with the course. The majority felt the content was mostly (10.6%) or very relevant (88.3%), and was somewhat (29.3%), mostly (34.0%) or very new (31.9%). Course content was rated as being either mostly (13.7%) or very culturally appropriate (86.2%) by participants who identified as Aboriginal or Torres Strait Islander. The majority of participants were very satisfied (88.8%) and 11.2% were mostly satisfied with the course overall, and a high percentage were very satisfied with the different course materials, including the handbook (79.7%), slides (75.4%), videos (88.2%), and activities (75.9%).

Just over one-third (70/188 = 37%) of participants reported feeling distressed about an aspect of the course. Of those people who experienced some level of distress, all but one indicated that they were glad to have attended the training despite this potential for distress. Just one person who experienced some level of distress indicated that they were "not sure" if they were glad to have attended the training.

## Discussion

This uncontrolled trial of the Talking About Suicide course yielded encouraging results, despite attrition in follow-up data and participants being largely 'experienced' in mental health in a variety of formal ways as well as having personal experience of suicidality. At post-course measurement, we observed improvements in beliefs about suicide, stigmatising attitudes, confidence in ability to assist and intended assisting behaviours. Generally, participants with the strongest, evidence-based responses maintained these. Participants who gave less sure or neutral responses frequently jumped to ceiling levels, and those with initially lower or non-evidence-based views showed positive responses after training. Participants reported feeling more confident to assist a suicidal person after the course, which was maintained at follow-up. Importantly, actual assisting actions taken showed dramatic improvements between pre-course and follow-up. Reactions to the course were very positive, with many participants responding that it contained information that was new to them. The course was judged to be culturally appropriate by participants who identified as Aboriginal or Torres Strait Islanders.

Attrition at follow-up was an issue impacting statistical power and potential representativeness of the remaining respondents, so follow-up findings should be interpreted in this light. Nonetheless, several improvements in beliefs about suicide, stigmatising attitudes and intended recommended assisting actions remained statistically significant at follow-up, and several other findings trended in a positive direction despite losing statistical significance.

Arguably, the most important marker of change in this study was the improvement in endorsement rates of intended assisting actions and in actual actions taken. The results in this area were especially encouraging and indicate that the course supported participants to identify and implement assisting actions that were consistent with the consensus guidelines [28]. Previous studies have observed that best practice intentions are highly correlated with best practice actions [42–44] and, consistent with this, our study identified substantial gains in both intended and actual assisting actions. Encouragingly, we also observed large effect sizes in some core intended and actual assisting actions. For example, there was a large improvement in both participants intended and actual assisting actions in relation to asking the person directly if they are experiencing suicidal thoughts and if they have a suicide plan, which aligned with a reduction in beliefs that asking about suicide can put the idea into someone's head. Consistent with the communication guidance provided in the training, which was underpinned by a culturally appropriate narrative or 'yarning' approach [45, 46], we also observed important improvements in intended or actual assisting actions in relation to spending time listening to the person and allowing them space to discuss their reasons for wanting to die.

We did observe a statistically significant increase in intentions to 'take charge of the situation', a non-recommended assisting action, which was mirrored by an increase, albeit non-significant, in participants reporting this as one of their actual assisting actions. The wording of this item in the survey may have been somewhat confusing for participants, and needs to be reviewed in future evaluations as it may reflect the increase in confidence to assist more than a real desire to "take charge" in a negative (i.e. disrespectful and paternalistic) way. Future evaluations might consider alternative phrasing, such as "take over", given that the course directly teaches participants not to take over and solve the problems for the person.

We also found an increase in intended or actual assisting actions in relation to trying to convince the person that suicide is the wrong thing to do, also a non-recommended assisting action, although the increases were not statistically significant. A review and refinement of the training materials may identify areas where these non-recommended assisting actions can be given greater attention.

## Study significance and course roll out

The Talking About Suicide course contributes to filling an important gap. There is a well-established need for the development of culturally appropriate suicide prevention initiatives for Aboriginal and Torres Strait Islander communities [47]. This evaluation provides encouraging findings, indicating that this suicide gatekeeper training course can support local community efforts in the area of Aboriginal and Torres Strait Islander suicide prevention. The Talking About Suicide course is relatively brief (i.e. ~5 hours in length) and in a short time period was able to demonstrate improvements in participant knowledge, attitudes and behaviours that were immediately and actively being used in communities to assist Aboriginal and Torres Strait Islander people experiencing suicidal thoughts. These findings are consistent with a systematic review of suicide interventions targeting Indigenous peoples globally, which found that gatekeeper training can produce improvements in participant knowledge, skills, intentions to assist, and confidence in identifying and assisting individuals at risk of suicide [16]. Our study extends the literature in this review by capturing data on changes in actual assisting behaviours, which has been something missing in prior research.

The program is yet to be fully advertised and rolled-out, allowing time for final adjustments to be made to the course based on the above-mentioned findings. Nonetheless, during the early implementation phase, 14 AMHFA Instructors have already been trained to deliver the

course and these instructors have delivered courses to 374 people (266 members of the public and 108 people in their workplace) across all states and territories in Australia.

## Limitations and future research

There are some study limitations to acknowledge. Firstly, the results of this trial may have been subject to a ceiling effect, given that approximately 50% had prior MHFA training and approximately 50% had engaged in other forms of mental health training. As a result, pre-course responses to most questions were generally strongly skewed to evidence-based positions. Nevertheless, there were clear and statistically significant improvements after the course, demonstrating that even participants who were generally quite skilled/knowledgeable due to prior MHFA training, still benefitted from the course. Secondly, participants were predominantly frontline workers with a substantial proportion having prior mental health training, making it difficult to generalise these findings to community members or those frontline workers without prior mental health training. Future research needs to investigate how well these results generalise to other populations that might be less experienced/knowledgeable in suicide prevention.

Thirdly, while our study was strengthened by the collection of follow-up data, four-months is a relatively short time period and there are no data on whether the positive effects of the training were maintained over longer time periods. In addition, our follow-up findings need to be interpreted cautiously due to a high rate of attrition. Individuals who identified as Aboriginal or Torres Strait Islander or who lived in remote areas, and participants who had less suicide prevention expertise based on pre-course responses may be under-represented in follow-up data. Fourthly, participants may have been influenced by social desirability bias, such that they reported responses to some of the questions that were more socially acceptable or were perceived to be desired by the training facilitators; this may have particularly impact responses to questions related to course satisfaction. Finally, our study design was weakened by the absence of a control group and some of the improvements observed may have been due to the effect of repeated measurements.

In terms of future research, there are several potential avenues of exploration including looking at a broader range of health-related outcome measures, for example, referral and treatment patterns and rates of suicide attempts. There is also value in extending understanding of cultural relevance beyond what we could derive from a single post-course question. It would be valuable to have a more nuanced understanding of the extent to which the cultural adaptation was acceptable, safe and relevant and the extent to which further cultural adaptations could be embedded in the course to extend its effectiveness and reach. Further research might also explore the extent to which non-Indigenous participants gained knowledge of contextual, social and cultural factors around suicide for Aboriginal and Torres Strait Islander people even though this was not an explicit learning objective. Finally, future AMHFA programs should consider as a matter of urgency recalibrating approaches to evaluation by giving Aboriginal and Torres Strait Islander researchers greater control of the research process, including determining evaluation methodologies.

## Conclusions

The results of this uncontrolled trial were encouraging, suggesting that over five hours the Talking About Suicide course was able to improve participants' attitudes and intended and actual assisting behaviours, including among those with prior experience and training. Assisting actions recommended during the training were implemented by many participants in the

four months since participating in the training course, demonstrating immediate benefits for the Aboriginal and Torres Strait Islander people who were experiencing suicidal thoughts.

## Acknowledgments

We have some sincere acknowledgments to make for several people. We would like to acknowledge the important role of the Aboriginal and Torres Strait Islander Project Reference Group, comprised of Professor Kerry Arabena, Ms Leilani Darwin, Mr Les Baird and Mr Jonathan Link. They provided essential cultural guidance and support throughout this study and the development of the training course. We would also like to acknowledge the participating Aboriginal Community-Controlled Organisations that were integral to both piloting the training course and implementing this evaluation study. We would like to acknowledge the authors of the training manual that formed the basis of the training course, Ms Eliza Pross, Dr Claire Kelly, Ms Fiona Blee and Ms Katrina Dart. We would like to thank the contributors to the lived experience films, which formed a critical element of the training materials, and the Instructors and study participants for sharing their time and passion for suicide prevention. We would also like to thank Mr Lewis Gould-Fensom for his contribution to data collection as a research assistant. Finally, we would like to thank Dr Betty Kitchener, Dr Claire Kelly, Professor Kerry Arabena and Dr Laura Hart for their contributions as investigators on the grant funding this study.

## Author Contributions

**Conceptualization:** Gregory Armstrong, Anthony F. Jorm.

**Data curation:** Georgina Sutherland, Eliza Pross.

**Formal analysis:** Gregory Armstrong, Andrew Mackinnon.

**Funding acquisition:** Nicola Reavley, Anthony F. Jorm.

**Investigation:** Gregory Armstrong, Georgina Sutherland, Anthony F. Jorm.

**Methodology:** Gregory Armstrong, Andrew Mackinnon, Anthony F. Jorm.

**Project administration:** Georgina Sutherland, Eliza Pross.

**Resources:** Eliza Pross.

**Supervision:** Anthony F. Jorm.

**Writing – original draft:** Gregory Armstrong, Anthony F. Jorm.

**Writing – review & editing:** Gregory Armstrong, Georgina Sutherland, Eliza Pross, Andrew Mackinnon, Nicola Reavley, Anthony F. Jorm.

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
