## [Decision Letter · Decision Letter 0]

10 Nov 2020

PONE-D-20-29679

Talking About Suicide: an uncontrolled trial of the effects of an Aboriginal and Torres Strait Islander Mental Health First Aid program on knowledge, attitudes and intended and actual assisting actions

PLOS ONE

Dear Dr. Armstrong,

Thank you for submitting your manuscript to PLOS ONE. After careful consideration, we feel that it has merit but does not fully meet PLOS ONE’s publication criteria as it currently stands. Therefore, we invite you to submit a revised version of the manuscript that addresses the points raised during the review process.

We look forward to receiving your revised manuscript.

Kind regards,

Danuta Wasserman

Academic Editor

PLOS ONE

Journal Requirements:

"We have some sincere acknowledgements to make for several people. We would like

to acknowledge the important role of the Aboriginal and Torres Strait Islander Project

Reference Group, comprised of Professor Kerry Arabena, Ms Leilani Darwin, Mr Les

Baird and Mr Jonathan Link. They provided essential cultural guidance and support

throughout this study and the development of the training course. We would also like

to acknowledge the participating Aboriginal Community-Controlled Organisations

that were integral to both piloting the training course and implementing this

evaluation study. We would like to acknowledge the authors of the training manual

that formed the basis of the training course, Ms Eliza Pross, Dr Claire Kelly, Ms

Fiona Blee and Ms Katrina Dart. We would like to thank the contributors to the lived

experience films, which formed a critical element of the training materials, and the

Instructors and study participants for sharing their time and passion for suicide

prevention. We would also like to thank Mr Lewis Gould-Fensom for his contribution

to data collection as a research assistant. The study was funded by a National Health

and Medical Research Council grant (#1076796) funded under the grant scheme

“Mental Health Targeted Call for Research into Suicide Prevention in Aboriginal and

Torres Strait Islander Youth”. We would like to thank Dr Betty Kitchener, Dr Claire

Kelly, Professor Kerry Arabena and Dr Laura Hart for their contributions as

investigators on the grant who are not authors on this paper. The lead author is funded

by an Early Career Fellowship (GNT1138096) from the National Health and Medical

Research Council in Australia.".

i) We note that you have provided funding information that is not currently declared in your Funding Statement. However, funding information should not appear in the Acknowledgments section or other areas of your manuscript. We will only publish funding information present in the Funding Statement section of the online submission form.

ii) Please remove any funding-related text from the manuscript and let us know how you would like to update your Funding Statement. Currently, your Funding Statement reads as follows:

iii) Please include your amended statements within your cover letter; we will change the online submission form on your behalf.

"AFJ is unpaid Chair of the Board of Mental Health First Aid Australia, which is a not-for-profit organization.".

i) Please confirm that this does not alter your adherence to all PLOS ONE policies on sharing data and materials, by including the following statement: "This does not alter our adherence to  PLOS ONE policies on sharing data and materials.” (as detailed online in our guide for authors http://journals.plos.org/plosone/s/competing-interests).  If there are restrictions on sharing of data and/or materials, please state these. Please note that we cannot proceed with consideration of your article until this information has been declared.

ii) Please include your updated Competing Interests statement in your cover letter; we will change the online submission form on your behalf.

4. Please include your tables as part of your main manuscript and remove the individual files. Please note that supplementary tables (should remain/ be uploaded) as separate "supporting information" files

Additional Editor Comments (if provided):

Please address the comments from the reviewers and revise the manuscript before resubmitting.

Reviewers' comments:

Reviewer's Responses to Questions

**Comments to the Author**

1. Is the manuscript technically sound, and do the data support the conclusions?

Reviewer #1: Yes

Reviewer #2: Yes

2. Has the statistical analysis been performed appropriately and rigorously? 

Reviewer #1: Yes

Reviewer #2: N/A

3. Have the authors made all data underlying the findings in their manuscript fully available?

Reviewer #1: Yes

Reviewer #2: Yes

4. Is the manuscript presented in an intelligible fashion and written in standard English?

Reviewer #1: Yes

Reviewer #2: Yes

5. Review Comments to the Author

Reviewer #1: Interesting study on a topic of general interest not only as regards suicide prevention in indigenous populations. It is not easy to evaluate suicide prevention programmes and this is an example of how it can be done.

It is also interesting and good that the authors present a rather short programme that has a chance to be used in a more long lasting suicide prevention programme which is needed. Suicide prevention is probably something that needs to be not too complicated if it should survive and not just be one of many "projects".

Table 4 and 5 should change place I think in the presentation.

Reviewer #2: Very interesting and important paper. However, the authors should write more in the discussion about the limitations in this study due to few subjects included and short follow-up. Recommendations for future research should also be added.

In the reference list, and cite somewhere in the text, please include the following references:

- Gergö Hadlaczky, Sebastian Hökby, Anahit Mkrtchian, Vladimir Carli & Danuta Wasserman (2014) Mental Health First Aid is an effective public health intervention for improving knowledge, attitudes, and behaviour: A meta-analysis, International Review of Psychiatry, 26:4, 467-475, DOI: 10.3109/09540261.2014.924910

- Zalsman G, Hawton K, Wasserman D et al. Suicide prevention strategies revisited: 10-year systematic review. Lancet Psychiatry. 2016; (published online June 8.) http://dx.doi.org/10.1016/S2215-0366(16)30030-X

- Wasserman D, Hoven CW, Wasserman C et al. School-based suicide prevention programmes: the SEYLE cluster-randomised, controlled trial. The Lancet, 385(9977), 1536-1544.

- Wasserman, C., Postuvan, V., Herta, D., Iosue, M., Värnik, P., & Carli, V. (2018). Interactions between youth and mental health professionals: The youth aware of mental health (YAM) program experience. PLoS ONE, 13(2). https://doi.org/10.1371/journal.pone.0191843

- Wasserman, C., Hoven, C. W., Wasserman, D., Carli, V., Sarchiapone, M., Al-Halabí, S., … Poštuvan, V. (2012). Suicide prevention for youth - A mental health awareness program: Lessons learned from the Saving and Empowering Young Lives in Europe (SEYLE) intervention study. BMC Public Health, 12(1). https://doi.org/10.1186/1471-2458-12-776

6. PLOS authors have the option to publish the peer review history of their article (what does this mean?). If published, this will include your full peer review and any attached files.

Reviewer #1: No

Reviewer #2: No

---

## [Author Response · Author response to Decision Letter 0]

15 Nov 2020

Response to reviewers and editorial requests

Editorial requests

1. “Funding information should not appear in the Acknowledgments section or other areas of your manuscript. We will only publish funding information present in the Funding Statement section of the online submission form.”

We have now removed the funding information from the acknowledgements section and moved it to the funding statement. Please find the new statements below: 

Acknowledgements:

We have some sincere acknowledgements to make for several people. We would like to acknowledge the important role of the Aboriginal and Torres Strait Islander Project Reference Group, comprised of Professor Kerry Arabena, Ms Leilani Darwin, Mr Les Baird and Mr Jonathan Link. They provided essential cultural guidance and support throughout this study and the development of the training course. We would also like to acknowledge the participating Aboriginal Community-Controlled Organisations that were integral to both piloting the training course and implementing this evaluation study. We would like to acknowledge the authors of the training manual that formed the basis of the training course, Ms Eliza Pross, Dr Claire Kelly, Ms Fiona Blee and Ms Katrina Dart. We would like to thank the contributors to the lived experience films, which formed a critical element of the training materials, and the Instructors and study participants for sharing their time and passion for suicide prevention. We would also like to thank Mr Lewis Gould-Fensom for his contribution to data collection as a research assistant. Finally, we would like to thank Dr Betty Kitchener, Dr Claire Kelly, Professor Kerry Arabena and Dr Laura Hart for their contributions as investigators on the grant funding this study.

Funding statement:

The study was funded by a National Health and Medical Research Council grant (#1076796) funded under the grant scheme “Mental Health Targeted Call for Research into Suicide Prevention in Aboriginal and Torres Strait Islander Youth”. The lead author is funded by an Early Career Fellowship (GNT1138096) from the National Health and Medical Research Council in Australia. The funders had no role in study design, data collection and analysis, decision to publish, or preparation of the manuscript.

2. “Please include your tables as part of your main manuscript and remove the individual files.”

We have now included the tables and figures in the main manuscript and removed the individual files.

3. Requested updates to competing interest statement

New competing interest statement:

AFJ is unpaid Chair of the Board of Mental Health First Aid Australia, which is a not-for-profit organization. This does not alter our adherence to PLOS ONE policies on sharing data and materials.

4. Title consistency

The correct title is:

Talking About Suicide: an uncontrolled trial of the effects of an Aboriginal and Torres Strait Islander Mental Health First Aid program on knowledge, attitudes and intended and actual assisting actions 

We have updated this in the online submission system.

Peer reviewer comments

Reviewer one

1. “Interesting study on a topic of general interest not only as regards suicide prevention in indigenous populations. It is not easy to evaluate suicide prevention programmes and this is an example of how it can be done. It is also interesting and good that the authors present a rather short programme that has a chance to be used in a more long lasting suicide prevention programme which is needed. Suicide prevention is probably something that needs to be not too complicated if it should survive and not just be one of many "projects". 

Thank you for these encouraging remarks.

2. “Table 4 and 5 should change place I think in the presentation.”

Without a rationale provided by the reviewer we are unclear as to how this would improve the presentation. We have elected to keep the current order of the tables, such that Table 4 still precedes Table 5. We understand this is an issue of style and there may be divergent views, but we feel it is appropriate to first focus on guideline concordant skills before focusing on guideline discordant skills.

Reviewer two

3. “Very interesting and important paper. However, the authors should write more in the discussion about the limitations in this study due to few subjects included and short follow-up. Recommendations for future research should also be added.”

The limitations section includes statements that address the issues caused by the short follow-up period and the sample size issues related to attrition (page 21), as per the text below:

‘Thirdly, while our study was strengthened by the collection of follow-up data, four-months is a relatively short time period and there are no data on whether the positive effects of the training were maintained over longer time periods. In addition, our follow-up findings need to be interpreted cautiously due to a high rate of attrition.’ 

Despite the issue of attrition, the sample size of 192 for pre/post data makes it one of the larger trials in Australia of a suicide prevention intervention for Aboriginal and Torres Strait Islander people.

4. “Recommendations for future research should also be added.”

The manuscript includes a paragraph (page 21-22) discussing recommendations for future research, as below:

‘In terms of future research, there are several potential avenues of exploration including looking at a broader range of health-related outcome measures, for example, referral and treatment patterns and rates of suicide attempts. There is also value in extending understanding of cultural relevance beyond what we could derive from a single post-course question. It would be valuable to have a more nuanced understanding of the extent to which the cultural adaptation was acceptable, safe and relevant and the extent to which further cultural adaptations could be embedded in the course to extend its effectiveness and reach. Further research might also explore the extent to which non-Indigenous participants gained knowledge of contextual, social and cultural factors around suicide for Aboriginal and Torres Strait Islander people even though this was not an explicit learning objective. Finally, future AMHFA programs should consider as a matter of urgency recalibrating approaches to evaluation by giving Aboriginal and Torres Strait Islander researchers greater control of the research process, including determining evaluation methodologies.’

5. “In the reference list, and cite somewhere in the text, please include the following references:

- Gergö Hadlaczky, Sebastian Hökby, Anahit Mkrtchian, Vladimir Carli & Danuta Wasserman (2014) Mental Health First Aid is an effective public health intervention for improving knowledge, attitudes, and behaviour: A meta-analysis, International Review of Psychiatry, 26:4, 467-475, DOI: 10.3109/09540261.2014.924910

- Zalsman G, Hawton K, Wasserman D et al. Suicide prevention strategies revisited: 10-year systematic review. Lancet Psychiatry. 2016; (published online June 8.) http://dx.doi.org/10.1016/S2215-0366(16)30030-X

- Wasserman D, Hoven CW, Wasserman C et al. School-based suicide prevention programmes: the SEYLE cluster-randomised, controlled trial. The Lancet, 385(9977), 1536-1544.

- Wasserman, C., Postuvan, V., Herta, D., Iosue, M., Värnik, P., & Carli, V. (2018). Interactions between youth and mental health professionals: The youth aware of mental health (YAM) program experience. PLoS ONE, 13(2). https://doi.org/10.1371/journal.pone.0191843

- Wasserman, C., Hoven, C. W., Wasserman, D., Carli, V., Sarchiapone, M., Al-Halabí, S., … Poštuvan, V. (2012). Suicide prevention for youth - A mental health awareness program: Lessons learned from the Saving and Empowering Young Lives in Europe (SEYLE) intervention study. BMC Public Health, 12(1). https://doi.org/10.1186/1471-2458-12-776”

Thank you for the suggested list of additional citations. We have reviewed them and included two; Hadlaczky et al 2014 and Zalsman et al 2016. The Hadlackzy et al 2014 article provides a meta-analysis of mental health first aid interventions. We have also inserted the reference below, which represents a more recent meta-analysis.

Morgan, A. J., A. Ross and N. J. Reavley (2018). "Systematic review and meta-analysis of Mental Health First Aid training: Effects on knowledge, stigma, and helping behaviour." PLoS One 13(5): e0197102.

The remaining suggestions were considered to be interesting studies although less directly relevant for inclusion in our manuscript.

---

## [Editor Report · Decision Letter 1]

3 Dec 2020

Talking About Suicide: an uncontrolled trial of the effects of an Aboriginal and Torres Strait Islander Mental Health First Aid program on knowledge, attitudes and intended and actual assisting actions

PONE-D-20-29679R1

Dear Dr. Armstrong,

We’re pleased to inform you that your manuscript has been judged scientifically suitable for publication and will be formally accepted for publication once it meets all outstanding technical requirements.

Kind regards,

Danuta Wasserman

Academic Editor

PLOS ONE
---

## [Editor Report · Acceptance letter]

7 Dec 2020

PONE-D-20-29679R1 

Talking About Suicide: an uncontrolled trial of the effects of an Aboriginal and Torres Strait Islander Mental Health First Aid program on knowledge, attitudes and intended and actual assisting actions 

Dear Dr. Armstrong:

I'm pleased to inform you that your manuscript has been deemed suitable for publication in PLOS ONE. Congratulations! Your manuscript is now with our production department. 

Kind regards, 

on behalf of

Dr. Danuta Wasserman 

Academic Editor

PLOS ONE